# Comparative Study of Terminal Cortical Potentials Using Iridium and Ag/AgCl Electrodes

**DOI:** 10.3390/ijms241310769

**Published:** 2023-06-28

**Authors:** Bulat Mingazov, Daria Vinokurova, Andrei Zakharov, Roustem Khazipov

**Affiliations:** 1Laboratory of Neurobiology, Kazan Federal University, Kazan 420008, Russia; BuRMingazov@kpfu.ru (B.M.); DaEVinokurova@kpfu.ru (D.V.); AnVZaharov@kpfu.ru (A.Z.); 2Department of Physiology, Kazan State Medical University, Kazan 420012, Russia; 3Institut de Neurobiologie de la Méditerranée (Inserm U1249), Aix-Marseille Université, 13273 Marseille, France

**Keywords:** neocortex, cortical spreading depolarization, negative ultraslow potential, respiratory arrest, global brain ischemia, general anesthesia, death

## Abstract

Brain ischemia induces slow voltage shifts in the cerebral cortex, including waves of spreading depolarization (SD) and negative ultraslow potentials (NUPs), which are considered as brain injury markers. However, different electrode materials and locations yield variable SD and NUP features. Here, we compared terminal cortical events during isoflurane or sevoflurane euthanasia using intracortical linear iridium electrode arrays and Ag/AgCl-based electrodes in the rat somatosensory cortex. Inhalation of anesthetics caused respiratory arrest, associated with hyperpolarization and followed by SD and NUP on both Ir and Ag electrodes. Ag-NUPs were bell shaped and waned within half an hour after death. Ir-NUPs were biphasic, with the early fast phase corresponding to Ag-NUP, and the late absent on Ag electrodes, phase of a progressive depolarizing voltage shift reaching −100 mV by two hours after death. In addition, late Ir-NUPs were more ample in the deep layers than at the cortical surface. Thus, intracortical Ag and Ir electrodes reliably assess early manifestations of terminal brain injury including hyperpolarization, SD and the early phase of NUP, while the late, giant amplitude phase of NUP, which is present only on Ir electrodes, is probably related to the sensitivity of Ir electrodes to a yet unidentified factor related to brain death.

## 1. Introduction

Electrophysiological recordings provide important insights into the mechanisms underlying the pathophysiology of ischemic brain damage [1,2,3,4,5,6,7,8]. Both in patients and in animal models, deprivation of metabolic supply causes a sequence of characteristic electrophysiological manifestations. These include suppression of physiological cortical activity and emergence of pathological patterns of spreading depolarizations (SD) [1,5,9,10,11,12,13,14,15,16] and SD-triggered negative ultralow potentials (NUP) [17,18,19,20,21,22]. Considerable evidence indicates that SDs and NUPs are reliable markers of ischemia and developing infarcts, which allows their operative monitoring and initiation of treatment [1,14,18]. NUPs are of particular interest because these events, discovered recently during the development of delayed focal cortical ischemia in patients with subarachnoid hemorrhage, as well as during terminal states using ECoG recording with Pt/Ir electrodes, can reach a giant amplitude exceeding −100 mV, which is the largest electric field potential ever recorded from the brain [17,18,19,20,21,22]. SD-triggered NUPs have also been demonstrated in animal models of hypoxia and global ischemia induced by cardiac arrest, as well as in endothelin-induced focal cortical ischemia [19,21]. Yet, previous studies also indicated a difference in NUP characteristics when recorded using electrodes made of noble materials (Pt/Ir or Ir) and Ag/AgCl-based electrodes [19,21]. In addition, the location of the electrodes relative to cortical depth may be an important factor in assessing SDs and NUPs, since both patterns may exhibit variable vertical organization profiles [21,23]. In the present study, we addressed these issues using simultaneous recording of extracellular field potential using linear arrays of Ir electrodes placed at different cortical depths and intracortical Ag/AgCl-based electrodes in rat somatosensory cortex using a terminal isoflurane or sevoflurane euthanasia model.

## 2. Results

We explored terminal voltage signals in the somatosensory cortex of adult head-restrained, urethane-anesthetized rats during euthanasia induced by inhalation of lethal doses of general volatile anesthetics isoflurane or sevoflurane (Figure 1A). Concomitant intracortical voltage recordings were performed using electrodes made of iridium (Ir), and silver/chlorided silver electrodes (Ag/AgCl, or Ag) as a standard for comparison. Ir electrodes were organized in linear 16-channel arrays (silicone probes), which also enabled characterizing the depth profile of the terminal voltage signals across cortical depth (Figure 1B).

### 2.1. General Characteristics of the Cascade of Terminal Events

Inhalation of lethal doses of isoflurane or sevoflurane caused a cascade of terminal events including respiratory and cardiac arrest as well as changes in electrical activity and extracellular voltage shifts in the cerebral cortex (Figure 2). The earliest manifestation of anesthetics’ effect was a rapid slowing-down of electrical activity with a transient emergence of the burst-suppression activity pattern, which is characteristic of high-dose isoflurane anesthesia [24,25,26,27]. This was followed by complete isoelectric silence (also known as “non-spreading depression” [4]) with cessation of electrical activity in the EEG frequency band (0.5–45 Hz) and complete suppression of MUA. The cessation of cortical electrical activity was accompanied by the development of a slow shift of the extracellular potential in the positive direction, hereafter referred to as hyperpolarization (HP). HP was followed by negative shift of the extracellular potential, heralded by a wave of SD and followed by NUP. While the suppression of EEG activity, the HP and SD developed on a relatively fast (minutes) time scale, NUPs were characterized by super-slow dynamics closer to the hourly range. Early voltage shifts, including HP, SD, and the NUP onset were qualitatively similar, although they differed to certain extent in amplitude on Ir and Ag electrodes. However, while NUPs were transient events that decreased in amplitude with time and disappeared on Ag electrodes, NUPs on Ir electrodes continued to develop for two hours after the animal’s death. Therefore, the term Ag-NUP was used to describe NUPs recorded with Ag electrodes, and the terms Ir-NUP1 and Ir-NUP2 were used to describe biphasic NUPs recorded with Ir electrodes.

During the preliminary analysis, it was found that the sequence of the events described above was reproduced in all animals; however, the time of their development relative to the beginning of the animals’ exposure to anesthetics varied considerably. At the same time, the temporal binding of the cascade of terminal events to the anesthetics-induced respiratory arrest (RA) was significantly higher. Therefore, further quantitative assessment of the dynamics of terminal events at the group level was performed in relation to RA, which was determined by the disappearance of respiratory movements, recorded by piezo detectors from the chest, occurring 145 (80–317) s (*n* = 32 rats) after the beginning of iso- or sevoflurane inhalation. The timing of key electrophysiological and vital terminal events relative to the time of respiratory arrest at the group level is shown in Figure 3. The onset of HP occurred approximately one minute before respiratory arrest (Ir: 55 (13–121) s (*n* = 21); Ag: 53 (20–137) s (*n* = 28)). In parallel with HP, isoelectric silence developed, the onset of which, determined by the time of the farewell activity burst in the cortex, occurred 31 (−3–86) s (*n* = 22) before RA. HP peaked 1 to 2 min after RA, and peak HP values were significantly higher on Ir electrodes (30 (18–35) mV (*n* = 22)) than on Ag electrodes (5 (3–7) mV (*n* = 28); *p* = 1 ×10^−5^). This was followed by the development of NUP, at the beginning of which a single SD wave was observed approximately 2 min after RA. ECG activity ceased 7.2 (5.6–9.1) min *(n* = 18) after RA.

### 2.2. Terminal SD

A single wave of terminal SD occurred 143 (115–187) s (*n* = 21) after RA on Ir electrodes and 144 (118–190) s (*n* = 28) after RA on Ag electrodes. Examples of simultaneous recordings of SD by a linear 16-channel array of Ir electrodes located at different depths of the barrel cortex and a single Ag/AgCl based electrode located at a depth of approximately 0.7 mm from the cortex surface, and at a distance of approximately 2 mm from the Ir probe obtained in three rats are shown in Figure 4. Despite a variety of SD depth profiles, terminal SD in all animals fall into the category of “full” SDs propagating through all cortical layers, which is typical for single SDs and the first SDs in clusters [21,23,28,29,30,31]. Within a single cortical column, in which a linear matrix of Ir electrodes was placed, the SD originated at a variable depth, and further propagated from this point toward the surface and deep into the cortex at a rate of approximately 2–5 mm/min (Figure 4A). On average, the point of terminal SD emergence was located at a depth of 0.5 (0.3–0.8) mm (*n* = 23) from the cortex surface, which is consistent with the increased propensity of the superficial cortical layers to SD in the various models [10,21,23,28,29,30,31,32]. However, there were also cases of terminal SDs propagating preferentially through the deep layers (Rat#1 from Figure 4A) [33]. The timing of SD on Ir and Ag electrodes (the Ir electrode, located at the depth of the Ag electrode (0.7 mm) was chosen for comparison) varied within time limits of ±tens of seconds in different animals, likely due to the horizontal spread of SD between horizontally spaced Ir and Ag electrodes. However, at the group level, this variability leveled off, and the time of occurrence of SD on Ir and Ag electrodes did not differ significantly (Figure 4B). Along with variability in the SD initiation depth, the vertical profile of the SD was also characterized by a greater SD amplitude in deeper layers of the cortex (Figure 4G). Further, the SD amplitude at Ir electrodes (−25 (−26–−19) mV at 0.7 mm depth (*n* = 16)) was approximately twice as large as on Ag electrodes (−12 (−14–−9) mV (*n* = 22; *p* = 0.0001).

### 2.3. Terminal NUP

The terminal NUP was ultra-slow negative potential shift developing on the hourly time scale (Figure 5). As noted above, NUPs on Ir and Ag electrodes differed not only quantitatively but also qualitatively: Ag-NUPs were transient events, whereas Ir-NUPs continued to develop for several hours after death and were organized into successive phases of Ir-NUP1 and 2. Ag-NUPs reached maximum amplitude of −12 (−18–−7) mV at a delay of 10.7 (7.2–13.3) min (*n* = 26) after RA, and then waned by 27.7 (18.0–82.6) min after RA. Until the peak of Ag-NUP, potential at the Ir electrodes followed similar trajectory, and the absolute potential values on the Ir electrodes at the time of the Ag-NUP peak (Ir-NUP: −13 (−32–−1) mV (*n* = 20) did not differ significantly from those on the Ag electrodes (*p* = 0.74; Figure 6). The biphasic Ir-NUPs were approximated by two exponents that corresponded to the Ir-NUP1 and Ir-NUP2 phases (Figure 5B): (1) a fast component with a relative contribution of 14 (1–35)% and a decay time constant of 3.7 (2.4–4.1) min, and (2) a slow component with a relative contribution of 86 (64–99)% and a decay time constant of 450 (95–8956) min (*n* = 24). The timepoint where the fast component decreases to 10% of its amplitude was considered as the Ir-NUP1/2 phase transition. The time difference between the Ir-NUP1/2 transition and the Ag-NUP peak (at 0.7 mm depth) was 95 (−78–221) s (*n* = 18, Figure 5C), and the Ir-NUP1/2 transition and Ag-NUP peak times differed little from each other (Figure 5D). The Ir-NUP1/2 transition occurred almost simultaneously at all cortical depths with only a slight tendency for the earlier occurrence of the Ir-NUP1/2 transition in the deep layers (Figure 5D). After the Ir-NUP1/2 transition and the Ag-NUP peak, the difference between the potential values at the Ir and Ag electrodes progressively increased with time (Figure 6). Whereas Ag-NUP waned by the end of a half an hour after RA, and the potential on the Ag electrodes after that time was close to zero, the potential on the Ir electrodes continued progressively shifting in the negative direction. By the end of the second hour after RA, the potential on Ir electrodes reached −81 (−93–−66) mV (*n* = 13), while the potential on Ag electrodes was around near-zero values of 2 (−4–14) mV (*n* = 16).

Potential values on the Ir electrodes across the cortical depth and on the Ag electrodes are shown in Figure 7. The potential values of Pt/Ir ECoG electrodes and intracortical Ag electrodes for some terminal reference timepoints from [19] are also shown for comparison. Analysis of the depth dependence revealed that the amplitude of HP was similar throughout the cortical depth, and, as noted earlier, the values of the HP peak on Ir electrodes were significantly higher than on Ag electrodes. Second, the negative potential values on the Ag electrodes at the Ag-NUP peak did not differ from those on the Ir electrodes throughout the cortex, and were also close to the values on the Pt/Ir ECoG electrodes [19]. Third, as slow NUP2 progressed on Ir electrodes, along with an increase in negative potential values at all depths, there was an increase in the vertical NUP gradient with greater NUP values in deep cortical layers. Indeed, the correlation coefficient (R) between potential and depth progressively increased from −0.23 during the Ag-NUP peak to −0.4 by 2 h after RA. Maximum Ir-NUP values of −97 (−119–−64) mV (*n* = 13) were attained by 2 h after RA at cortical depth of 1.4 mm. The tendency for further progression of Ir-NUP and its depth gradient persisted 2 h after RA, as evidenced by the negative slope of the potential values on Ir electrodes, and the greater value of this negative slope in the deep cortical layers (Figure 7).

## 3. Discussion

In the present study, we used intracortical Ir electrode arrays and Ag/AgCl-based electrodes to explore terminal voltage changes in the cerebral cortex of rats during death caused by volatile anesthetic overdose. Our main findings are twofold. Firstly, we found that early manifestations of terminal brain damage, including hyperpolarization, SD and the early phase of NUP, are reliably assessed with Ir and Ag electrodes. However, the late, giant amplitude phase of NUP was only present on Ir electrodes. Secondly, we found that the depth profiles of terminal SDs were variable, and that late Ir-NUPs were greater in the deep layers than on the surface of the cortex. On a methodological note, our results highlight the importance of taking into account the electrode materials and locations used to assess cortical electrical voltage variations during metabolic insults.

Cascade of terminal electrical potential changes in cerebral cortex during death involved a sequence of non-spreading depression, HP, SD and NUPs, which is consistent with results of previous studies in patients and animal models [19,20,34]. The transient pattern of burst suppression observed at the beginning of anesthetic inhalation, which has not been reported in other hypoxia/ischemia models, was probably specific to the isoflurane/sevoflurane model. However, the pattern of burst suppression failed to develop into the hypersynchronous epileptiform discharges characteristic of high but sublethal doses of these anesthetics [25,26,35]. This was probably due to a parallel development of non-spreading depression, an adaptive mechanism, which is aimed to reduce metabolic demand and to maintain vital functions such as ionic gradients and membrane potential, and which involves blockade of glutamate release via presynaptic adenosine receptors as a result of ATP degradation and elevated extracellular adenosine, and neuronal hyperpolarization caused by an increase in potassium conductance [1,36,37,38,39,40,41,42]. These potassium currents, together with isoflurane-activated two-pore potassium channels [25,43,44] are likely involved in the generation of HP. RA occurs shortly after the HP onset, as a result of depression of the respiratory brainstem center [45]. Interestingly, cardiac arrest was delayed from RA by 5–10 min, suggesting that the early changes in cortical potentials occurring during this period, including the HP, SD, and early NUP, were primarily caused by metabolic deficits resulting from hypoxia rather than ischemia.

Consistent with previous studies, SD, which is a hallmark cortical response to metabolic stroke [1,3,4], was also a characteristic event in the isoflurane overdose model in the present study despite of elevated threshold for SD induction reported under isoflurane anesthesia [46]. In contrast to focal ischemia, where multiple SDs are typically organized in clusters [14,21], only single SD wave was observed after RA that is in agreement with clinical findings and terminal models [19,20,34,47], as well as during severe metabolic insults in vitro [1,12,48]. Recordings across cortical depth revealed reach repertoire of vertical SD propagation profiles with the SDs more frequently starting in the superficial cortical layers as in other SD models, and propagating through all cortical layers thus corresponding to the “full” SD phenotype [10,21,23,28,29,30,31,32,49]. In addition, the terminal SDs were more ample in the deep layers of the cortex compared to the superficial layers, as were SDs induced by high potassium [23]. However, terminal SDs differed from SDs in the metabolically supplied cortex in two important aspects. First, while SDs are classically associated with spreading depression of cortical activity [9,50], terminal SDs occurred at the background of complete “isoelectric silence” caused by preceding non-spreading depression. Second, while “normoxic” SDs are relatively short events which last for approximately one minute and are curtailed by post-SD hyperpolarization, terminal SDs did not display such recovery and were followed by long-lasting NUP. These electrophysiological traits of the extracellular potential changes capture critical differences in the generative mechanisms underlying these two types of SD, which represent two extremes of the broad SD spectrum. At the cellular level, both “normoxic” and terminal SD are associated with almost complete neuronal depolarization and depolarization block of action potentials [1,51]. In metabolically supplied tissue, the maintained metabolism allows neurons to restore membrane potential primarily by boosting the electrogenic Na,K-APTase, which is associated with the outwardly directed transmembrane current contributing to the post-SD hyperpolarization. In contrast, terminal SDs (which are highly energy-consuming events) deplete the energy reserves, and in the absence of sources to maintain cell energy and to fuel Na,K-APTase, neurons fail to recover membrane potential and thus cannot restore membrane potential and thus remain permanently depolarized. The small transient positive shift in the extracellular potential that follows the SD peak (see examples in Figure 2B) may reflect activation of Na,K-APTase by residual energy stores. Thus, from the generative point of view, the early phase of NUP (Ag-NUP, and phase 1 of Ir-NUP) appears to be, in essence, the late phase of extremely prolonged terminal SD. This phase is a “commitment point”, during which neurons die if metabolic supply is not restored immediately [47,52,53].

The dynamics of early terminal events were basically similar in the recordings using Ir-based and Ag/AgCl-based electrodes. However, while Ag-NUPs peaked within ~10 min and disappeared half an hour after RA, Ir-NUPs showed a progressive increase and reached a maximum of ~−100 mV two hours after RA, by the time the potential on the Ag electrodes was close to zero. Late Ir-NUP2 were more than twice larger at the cortical depth than in the superficial layers that may explain relatively small size terminal NUPs reported in the rats using Pt/Ir ECoG electrodes [19] (see also Figure 7). Decomposition of biexponential Ir-NUPs revealed that the transition between the first fast component (Ir-NUP1) and the second slow component (Ir-NUP2) occurs closely to the peak of Ag-NUPs, and that Ir-NUP2 is specific for Ir electrodes. These observations and similar findings obtained in studies by Dreier and colleagues suggest that late, giant amplitude NUPs in terminal states, as well as in focal ischemia are not biological field potentials, but rather potentials generated at Ir (and Pt/Ir) electrodes as a result of chemical interferences including their oxygen and pH sensitivity [19,21,54]. Consistent with these results, NUP-like responses were also observed with Ir electrodes during focal kidney ischemia induced by endothelin-1 [55], suggesting that these pathological potentials are not brain-specific. Indeed, the Ir electrodes used in the present study have a O_2_ sensitivity of approximately 0.5 mV per 1% O_2_ [21], and assuming that interstitial O_2_ drops from 20% to 0% during the terminal state, the O_2_-dependent voltage shift at the Ir electrodes should be ~−10 mV. On the other hand, the Ir electrodes also exhibit a pH sensitivity of approximately −30 mV/pH, which according to the terminal pH shift from 7.3 to 6.7 [56] should result in a positive Ir electrode voltage response of ~15 mV. Therefore, some additional electrochemical processes at the Ir electrodes probably contribute to the giant, in the −100 mV range, NUPs. These may include polarization of Ir electrodes due to chemical reactions in the hydrous oxide layer, which can assume several oxidation states [57], caused by changes in the ionic composition of the extracellular space and products of biochemical reactions associated with neuronal death, including formation of “fixed negative charges”, leading to tissue edema [58,59,60]. In addition, although we heated the animals’ bodies during the experiment with a ventral thermal pad, respiratory and cardiac arrest probably cause the brain temperature to decrease to ambient temperature. Indeed, in rats euthanized by anesthetic overdose or after decapitation, the brain naturally cools down to room temperature approximately 25 min after cardiac arrest [61]. Because standard electrode potentials are temperature-dependent [62], this cooling of the brain may also influence NUPs. Nevertheless, NUPs recorded by the electrodes made of noble materials are highly indicative of infarct development. Therefore, identification of precise generative mechanisms of giant NUPs is a great challenge for further research, which can lead to optimization of this approach for operative detection and monitoring of ischemic cortical injury in patients.

## 4. Materials and Methods

Wistar rats of both sexes aged from 30 to 60 days were used. Animals were prepared under isoflurane anesthesia at the surgical level (4% for induction, 2% for maintenance, Fluotec4 (E-Z Systems Inc., Palmer, PA, USA), confirmed by a negative toe-pinch reflex. The skin above the skull was removed, Hemostab (Omegadent, Moscow, Russia) was used to stop capillary bleeding and then was rinsed with 0.9% NaCl. A metal ring was attached to the skull with dental cement (Grip Cement, Caulk Dentsply, DE, USA). Then isoflurane anesthesia was discontinued and the animals were administered urethane (1.5 g/kg, i.p.), and immobility, and the absence of vocalizations and negative toe-pinch reflexes were monitored during the entire experiment. During recordings, rats were placed on a heated platform (36 °C, TC-344B; Warner Instruments, Hamden, CT, USA). The metal ring was attached to a magnetic stand via a ball-joint to restrain head movements. Chloride-coated silver (Ag/AgCl) wire placed in the cerebellum served as a reference electrode. A cranial window 3–4 mm in diameter was drilled above the barrel cortex area (2.0 mm caudal and 5.5 mm lateral from bregma). The dura mater was cut in the area of electrode insertion. To avoid drying of the cortical surface the chamber was regularly supplied with warm artificial cerebrospinal fluid (ACSF) of the following composition: 126 mM NaCl, 3.5 mM KCl, 1.2 mM NaH_2_PO_4_, 25 mM NaHCO_3_, 20 mM glucose, 2 mM CaCl_2_, 1.3 mM MgCl_2_. Euthanasia was induced by inhalation of lethal doses of isoflurane (Karizoo, Barcelona, Spain) or sevoflurane (Abbott Laboratories, Chicago, IL, USA) applied using a cotton soaked in anesthetic placed in a conical plastic mask, which was brought to the animal’s nose (Figure 1).

Electrophysiological recordings of the field potential (FP) and multiple unit activity (MUA) were performed using linear 16-channel silicone probes with iridium electrodes: 413 μm^2^ recording site area, 100 μm separation distance (Neuronexus Technologies, Ann Arbor, MI, USA) and one channel Ag/AgCl wire electrode placed into a glass pipette electrode filled with ACSF. Glass pipette electrodes were pulled from borosilicate glass capillaries (GC150F-15, Clark Electromedical Instruments, Holliston, MA, USA). The Ir probe was inserted into the barrel cortex perpendicularly to the cortex surface to a depth of 1.6–1.8 mm. An Ag/AgCl pipette was inserted at an angle to Ir probe to achieve a minimal distance from the probe at depth 0.7 mm. The wideband signals (0–9 kHz) were amplified using a DigitalLynx (Neuralynx, Bozeman, MT, USA) amplifier set in the DC mode (input range ±131 mV) with partial suppression of DC and low-frequency signals by means of hardware RRC-filter [22], digitized at 32 kHz and saved on a PC for post hoc analysis using custom-written functions in MATLAB (MathWorks, Natick, MA, USA). The respiratory movements were recorded with a piezoelectric sensor placed under the chest. Electrocardiogram (ECG) was recorded with two entomological needles placed in front and back paw or in left and right front paw. For action potential detection, the raw wideband signal was filtered bandpass 300–4000 Hz and negative deflections exceeding 5 standard deviations were considered as spikes. The standard deviation was calculated over the most silent 1 s length segment of the filtered trace. Instantaneous MUA frequency was calculated in a sliding window 10 s wide with shifts of 1 s. FP was obtained from wideband signal by downsampling to 1000 Hz (resample function) and subsequent DC-reconstruction. FP power was calculated as the average spectral power of the signal in the range 0.5–45 Hz. Spectral analysis was performed by mean Chronux toolbox (mtspectrumc function). The DC-reconstruction was performed by calculating the inverse transfer function of the amplifier’s hardware filter with a preliminary assessment of the actual filter parameters through the analysis of outputs in response to the test signals injected to the amplifier input as described previously [22]. The reconstructed DC-FP signal was used to calculate parameters of HP, SDs and NUPs. Terminal SD parameters were calculated from the bandpass filtered signals in the 0.001–5 Hz range. Filtering with lowpass at 5 Hz was performed using Chebyshev type II digital filters (cheby2 function). Filtering with highpass at 1 mHz the digital RC-filter was used. The time of the maximal rate of depolarization at SD front was considered as SD onset [23]. To determine the parameters of HP and NUP, the DC signal was lowpass filtered at 1 Hz using digital RC-filter, followed by smoothing with sliding window of 4 s (smooth function). The HP onset was defined as the last timepoint when the first derivative of the processed FP signal was ≤0 before the HP peak. The two-exponential approximation of Ir-NUP was performed by Curve Fitting Toolbox (MathWorks). Ir-NUP1/2 transition was defined as a time of 90%-fall of the fast NUP component.

Statistical analysis was performed using the MATLAB Statistics toolbox. The rank sum test (ranksum function) was used for samples comparison. Correlations were calculated as Spearman’s correlation coefficient. Pooled data are shown as medians and 25th and 75th percentiles which are organized in boxplot or shaded area depicting the interquartile range and median line inside. The level of significance was kept at *p* < 0.05.

## Figures and Tables

**Figure 1 ijms-24-10769-f001:**
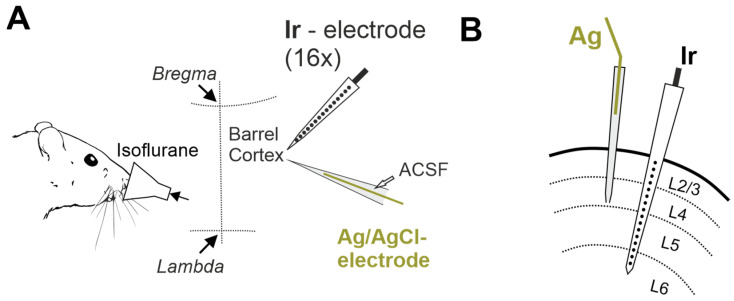
Experimental setup for recording terminal voltage signals in the rat cerebral cortex. (**A**) Head-restrained rats are exposed to lethal dose of volatile anesthetic isoflurane, and iridium (Ir) linear 16-electrode arrays and Ag/AgCl based electrodes via glass micropipette filled with ACSF are used to record voltage in the somatosensory barrel cortex. (**B**) Schematic drawing of the location of Ag/AgCl based (Ag) and Ir electrodes in the cerebral cortex on a coronal plane.

**Figure 2 ijms-24-10769-f002:**
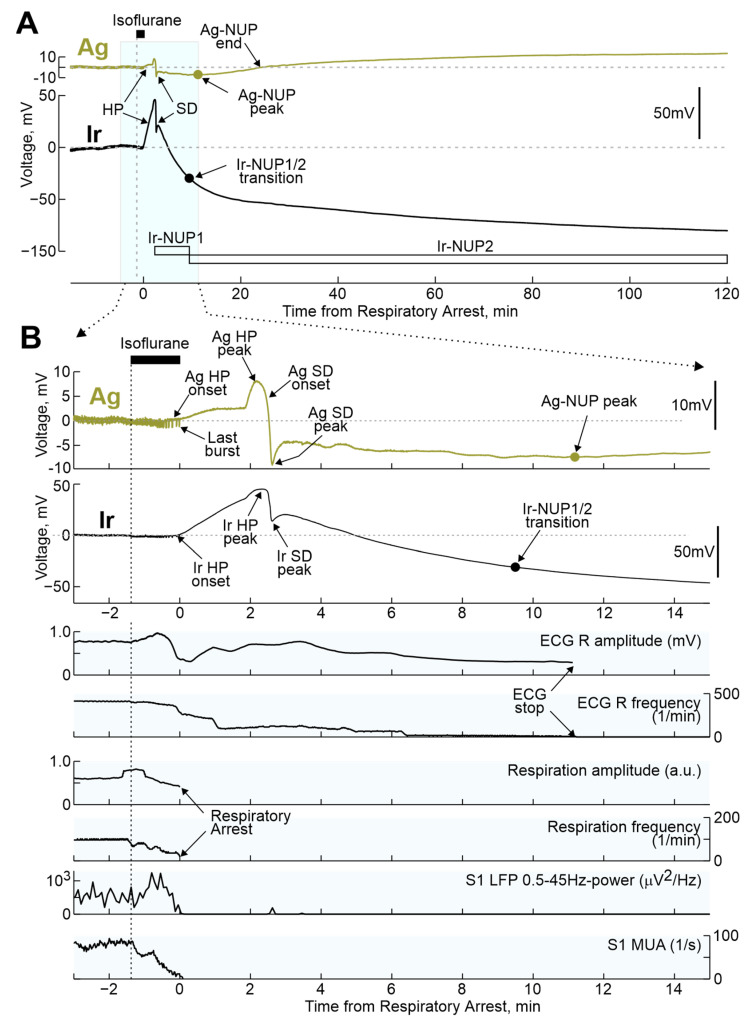
Terminal voltage changes in the rat cerebral cortex and vital functions during respiratory arrest induced by isoflurane inhalation. (**A**) Example of terminal voltage changes recorded in the rat cortex with Ir and Ag/AgCl electrodes at a depth of ~0.7 mm relative to the cortical surface during the period before and after respiratory arrest induced by isoflurane inhalation. (**B**) Fragment of recordings from Figure 1A, highlighted in blue, on an extended time scale. Below Ir- and Ag-electrograms, changes in the amplitude and frequency of R-peaks of electrocardiogram (ECG) and respiratory movements, power of LFP oscillations in the range of 0.5–45 Hz (Ir electrode) and MUA frequency at 0.7 mm cortical depth. HP, hyperpolarization; SD, spreading depolarization; NUP, negative ultraslow potential.

**Figure 3 ijms-24-10769-f003:**
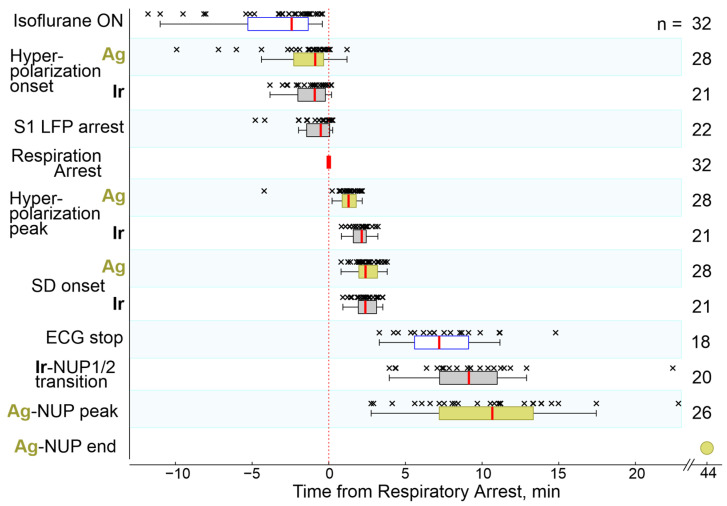
Timing of key electrophysiological and vital terminal events relative to respiratory arrest. The column on the right shows the number of animals used for an assessment of each parameter. Hereafter, each point (×) corresponds to an individual animal, boxplots show the median (red line) and interquartile (25–75%) range, whiskers indicate Q1 − 1.5 × (Q3 − Q1) and Q3 + 1.5 × (Q3 − Q1), where Q1 and Q3 are the 25th and 75th percentiles, respectively.

**Figure 4 ijms-24-10769-f004:**
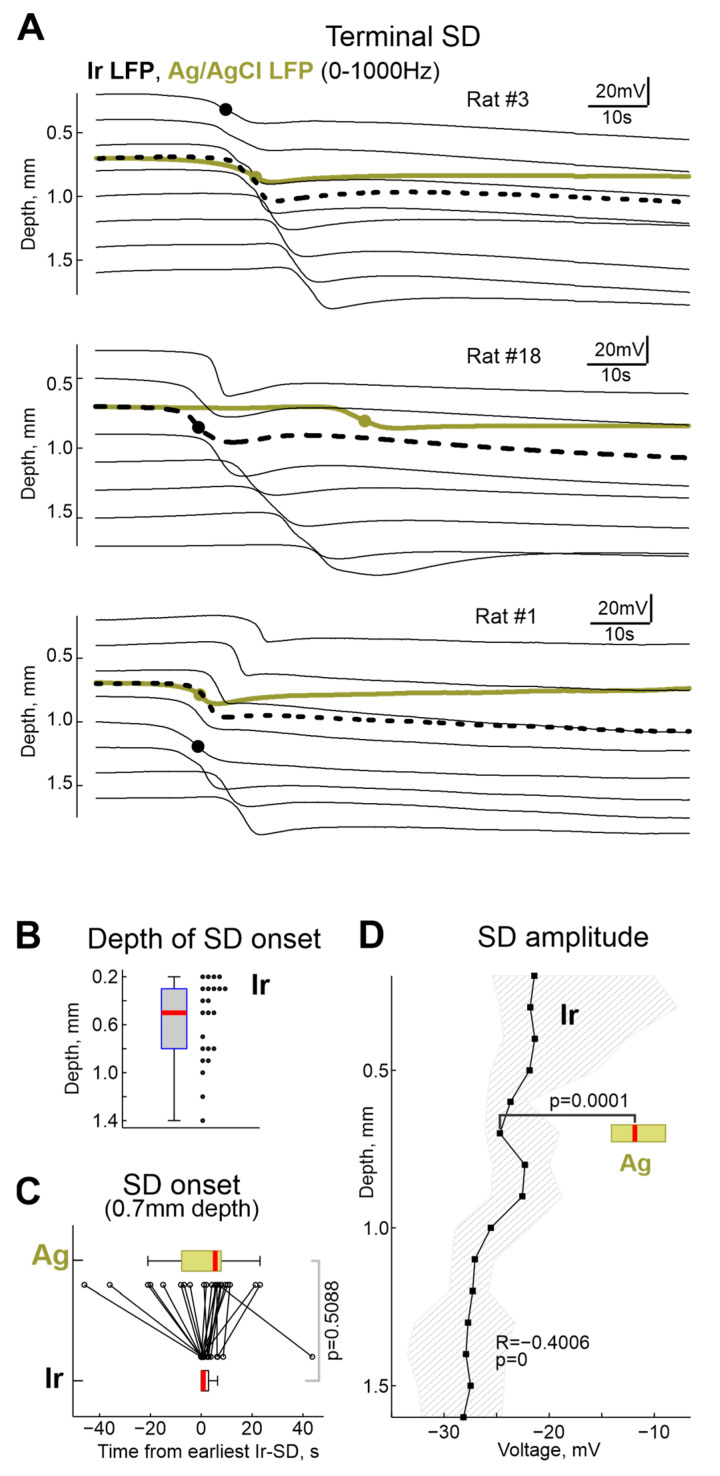
Terminal Spreading Depolarizations. (**A**) Examples of terminal SD in three rats. Black, electrode recordings at different cortical depth and olive, Ag/AgCl based electrode recordings at 0.7 mm depth. Black circles show the earliest onset of SD on Ir electrode array, olive circles show SD onset on Ag electrode. Black dashed lines correspond to the Ir electrode nearest to the Ag/AgCl based electrode. (**B**) Depth of terminal SD onset assessed with electrode arrays. (**C**) Comparison of SD onset on Ag/AgCl and Ir electrodes at 0.7 mm depth. Time = 0 corresponds to the earliest SD onset detected at the Ir electrode array. (**D**) Depth profile of terminal SD amplitude at Ir electrodes (median, shaded area indicates 25–75% percentiles range). The horizontal olive boxplot shows the distribution of the SD amplitudes on the Ag/AgCl electrode (at depth 0.7 mm). (**B**–**D**): pooled data from *n* = 23 rats.

**Figure 5 ijms-24-10769-f005:**
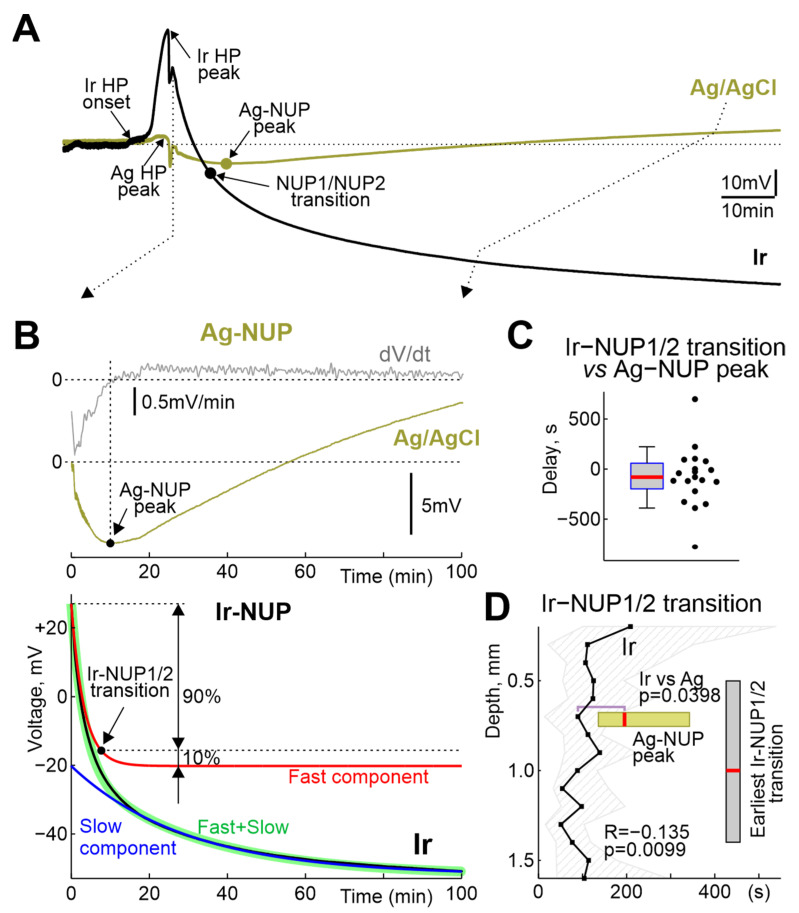
Comparative dynamics of NUPs at the Ir and Ag electrodes. (**A**) Example of the time course of the voltage during NUP on Ir and Ag/AgCl electrodes at a depth of 0.7 mm. (**B**) Fragment from the panel A, with the top traces showing raw (olive) and first derivative (grey) of the Ag-NUP, and bottom traces showing raw (black) Ir-NUP, its biexponential fit (green) and its fast (red) and slow (blue) components. Ir-NUP1/2 transition point (black circle) corresponds to the time when the fast component decays to 10% of its amplitude. (**C**) Distribution of the Ir-NUP1/2 transition times at 0.7 mm depth relative to the Ag-NUP peak. (**D**) Depth profile of the Ir-NUP1/2 transition times and the corresponding Ag-NUP peak times (horizontal olive boxplot). Zero time corresponds to the earliest Ir-NUP1/2 transition among all Ir array channels. Vertical grey boxplot shows distribution of the depth of the earliest Ir-NUP1/2 transition among Ir electrodes. (**C**,**D**): pooled data from *n* = 23 rats.

**Figure 6 ijms-24-10769-f006:**
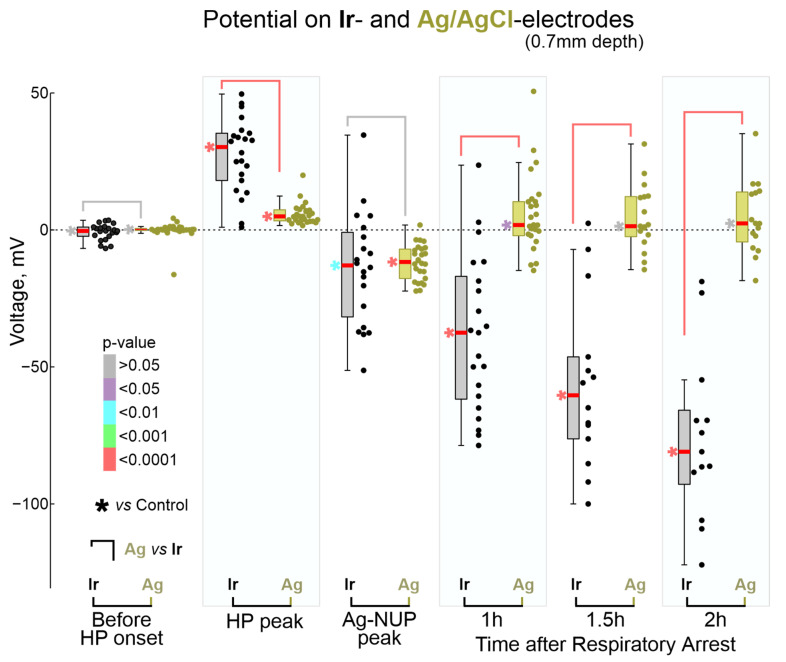
Comparison of absolute potential values on Ir and Ag/AgCl electrodes at the key moments of terminal ischemia at the cortical depth of 0.7 mm. Each dot corresponds to an individual animal (Ir—black, Ag—olive), * shows color-coded *p*-values for comparison with control values before anesthetic inhalation. Pooled data from *n* = 23 rats.

**Figure 7 ijms-24-10769-f007:**
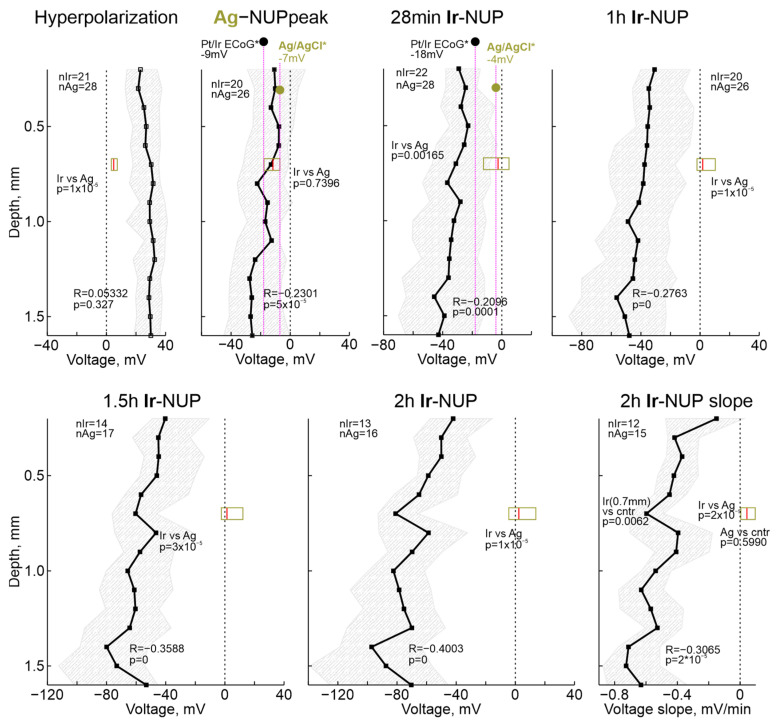
Depth profile of the absolute potential values on the Ir electrode array at several timepoints of terminal ischemia. Each dot shows median value, shaded area indicates 25–75% interquartile range. R, correlation coefficient; *p*, significance value for the correlation; *n*, number of animals. Horizontal boxplots show distribution of corresponding potential values on Ag electrodes. Black and olive circles show, for comparison, previously reported values for the corresponding times of global ischemia during Pt/Ir ECoG and intracortical Ag/AgCl based electrode recordings, respectively, in [19] (*). Right bottom, depth profile of the voltage slope of Ir-NUPs at the end of recordings (2 h).

## Data Availability

Original and processed data, and signal processing and analysis routines are available on request from the authors.

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
