# Peer review of "Comparative Study of Terminal Cortical Potentials Using Iridium and Ag/AgCl Electrodes"

_ijms, 2023, doi:10.3390/ijms241310769_

Round 1

Reviewer 1 Report

The authors studied the cascade of terminal electrical potential changes in the cerebral cortex during death involving a sequence of non-spreading depression, HP, SD, and NUPs, using intracortical Ir electrode arrays and Ag/AgCl-based electrodes. Statistical data were obtained on the time of development and the magnitude of the potentials as a function of the material of the electrodes used and the recording conditions (depth of electrode embedding). All results are perfectly illustrated and may be useful for researchers studying brain damage in stroke, trauma, etc. 

In principle, I have no comments and the article can be published as presented. However, the authors may be able to add a bit to the discussion by providing answers to the following questions:

1) A more detailed description of the electrochemical processes that may lead to NUPs.

2) Whether the body temperature of the rat was maintained after respiratory arrest. Could NUPs be influenced by changes in temperature?

3) How specific are NUPs to the cerebral cortex? Are these potentials observed in the white matter of the brain? In other organs?

Author Response

1) A more detailed description of the electrochemical processes that may lead to NUPs.

According to this advice, we added more detailed description of the factors contributing to late NUPs on Ir electrodes as follows (Pg. 12, Ln. 320-332; please see also response to the point 2):

Indeed, the Ir electrodes used in the present study have a O2 sensitivity of about 0.5 mV per 1% O2 [21], and assuming that interstitial O2 drops from 20% to 0% during the terminal state, the O2-dependent voltage shift at the Ir electrodes should be ~ -10 mV. On the other hand, the Ir electrodes also exhibit a pH sensitivity of about -30 mV/pH, which according to the terminal pH shift from 7.3 to 6.7 [56] should result in a positive Ir electrode voltage response of ~ 15 mV. Therefore, some additional electrochemical processes at the Ir electrodes probably contribute to the giant, in the -100 mV range, NUPs. These may include polarization of Ir electrodes due to chemical reactions in the hydrous oxide layer, which can assume several oxidation states [57], caused by changes in the ionic composition of the extracellular space and products of biochemical reactions associated with neuronal death, including formation of "fixed negative charges", leading to tissue edema [58-60].

2) Whether the body temperature of the rat was maintained after respiratory arrest. Could NUPs be influenced by changes in temperature?

This important point by the reviewer has been addressed in the discussion of the revised manuscript as follows (Pg. 12, Ln. 332-337):

In addition, although we heated the animals' bodies during the experiment with a ventral thermal pad, respiratory and cardiac arrest probably causes the brain temperature to decrease to ambient temperature. Indeed, in rats euthanized by anesthetic overdose or after decapitation, the brain naturally cools down to room temperature about 25 minutes after cardiac arrest [61]. Because standard electrode potentials are temperature-dependent [62], this cooling of the brain may also influence NUPs. 

3) How specific are NUPs to the cerebral cortex? Are these potentials observed in the white matter of the brain? In other organs?

We thank the reviewer for this excellent question. While we have no data to report about the white matter, in a preliminary study we have observed NUP-like potentials in the ET1-model of kidney ischemia. This information is added to Discussion as follows (Pg. 12, Ln. 318-320):

Consistent with these results, NUP-like responses were also observed with Ir electrodes during focal kidney ischemia induced by endothelin-1 [55], suggesting that these pathological potentials are not brain-specific.

Reviewer 2 Report

Dear authors

The paper "Comparative Study of Terminal Cortical Potentials using Iridium and Ag/AgCl electrodes" by Mingazov et al. was revised as requested previously and the manuscript fulfils the high standards required for publication in “International Journal of Molecular Sciences”.

We analyzed the originality, scientific quality, relevance to the field, presentation and adequacy of the references of the paper.

This manuscript is acceptable after minor revision:

- English language and style are fine/minor spell check required).

- Include more recent references (2023) on the subject.

With kind regards,

- English language and style are fine/minor spell check required).

Author Response

 - English language and style are fine/minor spell check required).

The paper has been checked for English

- Include more recent references (2023) on the subject.

Five more recent references (2023) have been included to the paper:

  1. Torteli, A.; Toth, R.; Berger, S.; Samardzic, S.; Bari, F.; Menyhart, A.; Farkas, E., Spreading depolarization causes reperfusion failure after cerebral ischemia. Journal of Cerebral Blood Flow and Metabolism 2023, 43, (5), 655-664.
  2. Meinert, F.; Lemale, C. L. G.; Major, S.; Helgers, S. O. A.; Domer, P.; Mencke, R.; Bergold, M. N.; Dreier, J. P.; Hecht, N.; Woitzik, J., Less-invasive subdural electrocorticography for investigation of spreading depolarizations in patients with subarachnoid hemorrhage. Front. Neurol. 2023, 13.
  3. Horst, V.; Kola, V.; Lemale, C. L.; Major, S.; Winkler, M. K. L.; Hecht, N.; Santos, E.; Platz, J.; Sakowitz, O. W.; Vatter, H.; Dohmen, C.; Scheel, M.; Vajkoczy, P.; Hartings, J. A.; Woitzik, J.; Martus, P.; Dreier, J. P., Spreading depolarization and angiographic spasm are separate mediators of delayed infarcts. Brain Communications 2023, 5, (2).
  4. Giniatullin, R.; Khazipov, R.; van den Maagdenberg, A.; Jolkkonen, J., Common and distinct mechanisms of migraine and stroke. Frontiers in Cellular Neuroscience 2023, 17.
  5. Alsbrook, D. L.; Di Napoli, M.; Bhatia, K.; Desai, M.; Hinduja, A.; Rubinos, C. A.; Mansueto, G.; Singh, P.; Domeniconi, G. G.; Ikram, A.; Sabbagh, S. Y.; Divani, A. A., Pathophysiology of Early Brain Injury and Its Association with Delayed Cerebral Ischemia in Aneurysmal Subarachnoid Hemorrhage: A Review of Current Literature. Journal of Clinical Medicine 2023, 12, (3).